# Maternal Origins and Haplotype Diversity of Seven Russian Goat Populations Based on the D-loop Sequence Variability

**DOI:** 10.3390/ani10091603

**Published:** 2020-09-09

**Authors:** Tatiana Deniskova, Nekruz Bakoev, Arsen Dotsev, Marina Selionova, Natalia Zinovieva

**Affiliations:** 1L.K. Ernst Federal Research Center for Animal Husbandry, Dubrovitzy Estate, Podolsk District, Moscow Region, 142132 Podolsk, Russia; nekruz82@bk.ru (N.B.); arsendotsev@vij.ru (A.D.); n_zinovieva@mail.ru (N.Z.); 2Russian State Agrarian University—Moscow Timiryazev Agricultural Academy, 127550 Moscow, Russia; m_selin@mail.ru

**Keywords:** goat, livestock, mtDNA, local breeds, haplogroup, phylogeny

## Abstract

**Simple Summary:**

Russia has diverse specifically selected and multipurpose goat resources. However, the origin of the local goats is still enigmatic. In this study, we sequenced and analyzed mitochondrial DNA (mtDNA) fragments of seven Russian local goat populations to provide the first insight into their maternal lineage.

**Abstract:**

The territory of modern Russia lies on the crossroads of East and West and covers various geographical environments where diverse groups of local goats originated. In this work, we present the first study on the maternal origin of Russian local goats, including Altai Mountain (*n* = 9), Dagestan Downy (*n* = 18), Dagestan Local (*n* = 12), Dagestan Milk (*n* = 15), Karachaev (*n* = 21), Orenburg (*n* = 10), and Soviet Mohair (*n* = 7) breeds, based on 715 bp D-loop mitochondrial DNA (mtDNA) sequences. Saanen goats (*n* = 5) were used for comparison. Our findings reveal a high haplotype (HD = 0.843–1.000) and nucleotide diversity (π = 0.0112–0.0261). A total of 59 haplotypes were determined in the Russian goat breeds, in which all differed from the haplotypes of the Saanen goats. The haplotypes identified in Altai Mountain, Orenburg, Soviet Mohair, and Saanen goats were breed specific. Most haplotypes (56 of 59) were clustered together with samples belonging to haplogroup A, which was in accordance with the global genetic pattern of maternal origin seen in most goats worldwide. The haplotypes that were grouped together with rare haplogroups D and G were found in the Altai Mountain breed and haplogroup C was detected in the Soviet Mohair breed. Thus, our findings revealed that local goats might have been brought to Russia via various migration routes. In addition, haplotype sharing was found in aboriginal goat populations from overlapping regions, which might be useful information for their official recognition status.

## 1. Introduction

Russia has a rich supply of goats, including officially recognized local (Altai Mountain, Dagestan Downy, Orenburg Down, and Soviet Mohair) and cosmopolitan breeds (Saanen, Nubian, Murciana Granadina) [1]. In addition, indigenous populations have long been reared in various regions [2]. Considering the predominantly harsh climate of Russia, goat breeding for down/cashmere and mohair/fur has a long national tradition, as well as economic importance, for providing Russians with warm winter clothing (scarfs, coats, sweaters). In this respect, the population of mohair and downy goats was the highest at the end of 1991, when it included 141,900, 111,700, 109,900, and 9800 Soviet Mohair, Orenburg, Altai Mountain, and Dagestan Downy goats, respectively [3]. Altai Mountain, Orenburg, and Soviet Mohair breeds were developed in regions with sharp continental environments, namely the Altai Mountains, Ural, Baikal, and Kazakhstan areas, respectively (Figure 1). On the contrary, the Dagestan Downy breed was obtained in the moderately continental and arid Southern Russia zone.

The consequences of the severe economic collapse led to a significant dramatic decline in the number of goats. In addition, the popularity and cheaper price of synthetic fibers changed the breed proportions in the Russian goat industry. Thus, the contemporary population at the end of 2019 was 28,600, 10,800, and 6500 goats for Soviet Mohair, Altai Mountain, and Orenburg breeds, respectively [1,4]. The latest official records for the Dagestan Downy breed (19,500 heads) were in 2010 [1].

On the contrary, dairy goats have been spreading widely due to the increasing demand for non-allergic food products and the possibility of rearing on large industrial farms. Among the dairy breeds, Saanen has the highest population, which went from 500 heads in 2004 to 29,770 heads by the end of 2019 [1,4].

Indigenous goat populations hold special importance because they meet the needs of locals for milk to produce cheese, and they provide the cheapest meat in mountainous and hard-to-reach areas with scarce forage resources, where breeding of other farm animals is difficult or economically impractical [5,6]. Because the Dagestan Milk and Karachaev populations have not been officially recognized as breeds yet, there are no official recordings of their population size. The numbers of the Dagestan Local breed increased from 9800 heads in 1991 [3] to 19,500 heads in 2010 [1]. However, according to personal communication, there are 5000 heads of Dagestan Downy, 100,000–110,000 heads of Dagestan Local, 6000 heads of Dagestan Milk, and 8000 heads of Karachaev breeds.

Polymorphisms of mitochondrial DNA (mtDNA) are widely used to clarify the origin, to study the processes of domestication, and to address migration routes of domestic goats and their wild relatives, as well as to establish familiar relationships via maternal lineages [7]. To address such important genetic questions, different regions of mtDNA have been sequenced and analyzed, including the hypervariable region HV1 [8], D-loop [9,10,11], cytochrome B [12], protein coding genes [13], and the complete mitogenomes [7,14].

Russian goat breeds mainly originated from native nondescript goats crossed with males from highly productive breeds [15,16], which were Angora for Soviet Mohair, local Don for Altai Mountain, and Soviet Mohair for the Dagestan Downy breed. The Orenburg breed is the result of the long-term improvement in the quality and color of the down fibers of native goats. However, many aspects of the origin of Russian local goats remain unexplored and enigmatic [15,16]. Furthermore, it is unknown how and when indigenous goats inhabited their domestic regions. In addition, establishment of the phylogenetic relationships of the goat populations indirectly indicates the migration routes of the Eurasian ethnic groups [17], which characterizes important social and cultural aspects of the study. Thus, analysis of mtDNA sequences will allow us to address the maternal origins, genetic diversity, and phylogenetic links of Russian local goat populations.

In addition, local Russian breeds are not monitored by DNA markers that might have negative consequences on their future status. In this regard, genetic characterization based on identification of the maternal lineage will contribute to the design of conservation programs by establishment of nuclear families to reproduce specific valuable or rare haplotypes.

In this research, we aimed to study genetic diversity and determine haplotype variability and haplogroup membership of the Russian local goats based on D-loop mtDNA sequences.

## 2. Materials and Methods

### 2.1. Sample Collection

A total of 97 goat samples were collected for this study. The sample included representatives of officially recognized local Russian breeds, such as Altai Mountain (ALTM, *n* = 9), Dagestan Downy (DAGD, *n* = 18), Orenburg (OREN, *n* = 10), and Soviet Mohair (SOVM, *n* = 7), and indigenous goat populations, including Dagestan Local (DAGL, *n* = 12), Dagestan Milk (DAGM, *n* = 15), and Karachaev (KARA, *n* = 21). Saanen goats (SAAN, *n* = 5) were used as a comparison group. The study was performed in accordance with the ethical guidelines of the L.K. Ernst Federal Research Center for Animal Husbandry. Protocol No 3/1 was approved by the Commission on the Ethics of Animal Experiments of the L.K. Ernst Federal Research Center for Animal Husbandry on 4 December 2019. Goat ear pinches were collected by trained personnel under strict veterinary rules in accordance with the rules for conducting laboratory research (tests) in the implementation of the veterinary control (supervision) approved by Council Decision Eurasian Economic Commission № 80 (November 10, 2017). We selected animals that were phenotypically typical for each breed. Only unrelated goats were used in the present study. The relationships were checked by the algorithms implemented in ML-Relate [18] based on preliminary STR-genotyping data. Specimens of Altai Mountain, Orenburg, and Soviet Mohair breeds were collected in relevant officially registered breeding farms. Samples of Dagestan Downy, Dagestan Local, Dagestan Milk, and Karachaev populations were obtained from smallholders. Saanen samples were taken from a breeding farm, which initially imported sires from France.

### 2.2. PCR Amplification, Purification, and Sequencing

Genomic DNA was extracted from ear pinches using commercial DNA-Extran-2 kits (CJSC Syntol, Russia) according to the manufacturer’s instructions. The mtDNA D-loop sequence, which was obtained from the National Center for Biotechnology Information (- (Gene Bank accession number NC_005044.2, a total length 16,643 bp), was used as a reference to design a pair of primers (Forward: 5′-ATACCAGCAGCTAGCACCATT-3′ and Reverse: 3′-GGCATTTTCAGTGCCTTGCTT-5′) to amplify the D-loop region of 1437 bp. The total volume for PCR amplification was 25 µL, which included 3 µL of DNA template (75 ng/µL), 5 µL 10 × PCR-buffer, 1 µL dNTP (1 mmol/L), 0.5 µL of each primer (20 pmol) and 0.5 µL of Smart Taq DNA polymerase (Dialat ltd, Moscow, Russia). The amplification was performed on a TProfessional Standard Gradient Thermocycler (Biometra, Germany) under the following conditions: 94 °C for 4 min; 33 cycles of 95 °C for 45 s, 65 °C for 45 s, and 72 °C for 45 s; and a final extension at 72 °C for 7 min. The quality of PCR product was checked by 1.5% agarose gel electrophoresis with ethidium bromide (0.5 μg/mL). Single well-defined bands were cut out from agarose gel with a surgical scalpel and were purified using the Cleanup Standard kit for the isolation and purification of DNA from agarose gel and reaction mixtures (JSC Eurogene, Moscow, Russia). An amplicon concentration was determined using a Qubit 3.0 fluorometer (Thermo Fisher Scientific (formerly Life Technologies), Wilmington, DE, USA). Purified PCR products were submitted to JSC Eurogene (Moscow, Russia) to sequence by Sanger technology from both ends.

### 2.3. Sequence Alignment and Data Analysis

Editing and alignment of the nucleotide sequences was performed using the BioEdit v7.2.6 [19] and MEGA 7 [20] software. The number of haplotypes (H), haplotype diversity (HD), nucleotide diversity (π), number of variable sites (S), and the average number of nucleotide differences (k) were calculated using DnaSP 5.10 [21]. To find the source of genetic variation among populations, analysis of molecular variance (AMOVA) was performed in Arlequin v3.5 [22]. The median-joining network was constructed in the PopART software [23] with prior MUSCLE alignment [24] implemented in the R package msa [25]. Determination of the best models of evolution was carried out in the program PartitionFinder 2 [26] using the Akaike information corrected criterion (AICc) [27]. The most optimal was the evolutionary model GTR+I+G. The construction of the Bayesian phylogenetic tree was carried out using the program MrBayes 3.2.6 [28] with subsequent visualization in FigTree 1.4.2 [29]. To establish the genetic relationships of the studied goat populations with the goats of different geographical locations and with wild bezoars, we additionally obtained the 16 sequences of the D-loop of mtDNA from the NCBI database (Appendix A). The D-loop sequence of Caucasian tur (*Capra caucasica*) (Gene Bank accession number NC__020683) was used as the outgroup. The Markov chain Monte Carlo search was run with four chains for 10,000,000 generations, with trees being sampled every 500 generations (the first 25% of trees were discarded as ‘burnin’). The geographic map (with longitude and latitude coordinates for each sampling site) was plotted using the R packages maps and mapdata [30].

## 3. Results

### 3.1. D-loop MtDNA Sequence Variation and Genetic Diversity

In this research, we investigated the variability of 97 D-loop mtDNA fragments spanning from 15,436 to 16,153 bp from seven Russian local goat populations and from the Saanen breed, which is popular in Russia, to address their phylogenetic links, haplotype and genetic diversity.

A nucleotide sequence analysis revealed that the number of variable sites ranged from 28 in the Dagestan Local to 57 in the Soviet Mohair breeds (Table 1). By the numbers of variable sites, all Russian goat groups exceeded the Saanen sample, including Altai Mountain and Soviet Mohair breeds with the closest sample numbers.

The largest average numbers of nucleotide differences were found in the Soviet Mohair (k = 18.667) and Altai Mountain breeds (k = 16.222). The k-value of the Dagestan Milk goats was close to those found in the Saanen breed (k = 10.781 and k = 10.600, respectively). The Dagestan Local, Karachaev, and Orenburg groups had the lowest numbers of nucleotide differences among Russian goats, as well as in comparison with the Saanen sample.

Altai Mountain, Soviet Mohair, Dagestan Downy, and Dagestan Milk breeds were characterized by a higher nucleotide diversity (π values varied from 0.01542 to 0.02607), while Orenburg, Karachaev, and Dagestan Local groups had lower π values (from 0.01027 to 0.01234) than Saanen goats.

Haplotype diversity was the highest in the Orenburg, Soviet Mohair, and Saanen breeds and was equal to 1.000 in each listed breed. In our sample, the Karachaev goat population had the lowest haplotype diversity (HD = 0.843).

AMOVA analysis showed that 77.24% of the total genetic variation of Russian goats occurred within breeds, and 22.76% of the variation was due to genetic differences among populations (Table 2). Considering only Russian goats, which were clustered with samples carrying haplogroup A, we found that there were no significant changes. Thus, 75.02% of the total genetic variation was accounted for by differences within breeds and 24.98% occurred among breeds.

### 3.2. Analysis of Phylogenetic Links and Haplogroup Assignment of Russian Local Goats

A total of 59 haplotypes were determined in Russian goat breeds. All identified haplotypes differed from the haplotypes found in Saanen goats (Table 3, Appendix A). The haplotypes identified in Altai Mountain, Orenburg, and Soviet Mohair and Saanen goats were breed specific. Six identified haplotypes were not breed specific. Thus, haplotypes H-11, H-18, and H-20 were shared by Dagestan Downy and Dagestan Local breeds; haplotype H-16 was shared by Dagestan Downy and Dagestan Milk groups; haplotypes H-28 and H-33 were shared by the Dagestan Milk and Karachaev groups.

Median-joining networks performed for the 64 various haplotypes, including 59 haplotypes of Russian and five haplotypes of Saanen goats, revealed formation of two clusters (Figure 2). The first included the majority of Russian goats, as well as all Saanen goats. The second comprised one Soviet Mohair animal and two Altai Mountain animals. The Soviet Mohair individual was separated from the closest nod with the first cluster by 44 mutations, Altai Mountain goats differed at the same position by 19 and 31 mutations, respectively.

To study the genetic relationships of Russian local goats, we obtained 16 reference sequences, presenting six major haplogroups in domestic and wild goats, to construct a phylogenetic tree (Figure 3). We found that 95% (56 of 59) of the studied haplotypes were clustered with samples belonging to haplogroup A. The haplotypes that were grouped together with rare haplogroups D and G were found in the Altai Mountain breed. In addition, a Soviet Mohair individual was clustered with representatives of haplogroup C.

## 4. Discussion

Goats are a popular livestock species in Russia and are reared in diverse environments including the Ural and Siberian parts with a harsh continental climate (Altai Mountain, Orenburg, and Soviet Mohair breeds) [15], as well as humid high altitudes in the North Caucasus Range (all Dagestan and Karachaev groups) [5,6,15,16]. Genetic investigations in the goat populations bred in Russia have been limited to application of nuclear DNA markers, such as microsatellites [31] and Single Nucleotide Polymorphisms (SNP-markers) [32]. In this regard, this study is the initial step to understand their maternal origin and to evaluate genetic diversity based on D-loop mtDNA sequences in Russian native goats.

The distribution of haplotypes in seven Russian goat groups partially reflects their geo-evolutive relationships and probably corresponds to their established status as breeds. Three officially recognized breeds, Altai Mountain, Orenburg, and Soviet Mohair, are characterized by the presence of exclusively breed-specific haplotypes. Each breed is reared in a particular region separated by significant geographical distances (about 2000 km between each one pairwise). Thus, for example, the Orenburg breed might be considered as an “endemic” breed to the Orenburg region because the repeated attempts to acclimatize these goats and to obtain their high quality down fibers in other regions failed [33]. A similar pattern might be found in relation to Altai Mountain and Soviet Mohair breeds, in which samples were collected in the Altai Region and Republic of Tyva, respectively.

On the contrary, goat groups that have shared haplotypes inhabit small republics situated in the North Caucasus region and are close geographically. Dagestan Local is an indigenous unimproved breed [34,35], which was used to obtain the improved Dagestan Downy and Dagestan Milk goat groups. Despite Dagestan Local and Dagestan Downy goat groups being considered as breeds, current breeding management is probably not sufficient for these goats as there is no official recording of breeding valuation. The genetic relations of Karachaev and Dagestan goats have not been established yet. However, the geographical proximity might result in shared haplotypes. Official recognition of Karachaev and Dagestan Milk goat populations is still in progress, and the choice of animals belonging to unadmixed maternal lineages to create nucleus families for further multiplying might facilitate the process.

As this is the first study on the mtDNA sequences of Russian goats, we compared calculated genetic diversity indicators with those estimated in various breeds to understand general genetic patterns.

For this study, we collected larger sample numbers from each population (from 20 to 25 samples) than the numbers that were submitted for further sequencing of the D-loop. This was because there were a lot of highly-related animals in our initial collection that would create possible biases in estimation of genetic diversity indicators. Thus, one animal from a pair of full-sibs was left in our analysis. Haplotype diversity in Russian goat populations was similar to the values obtained in various studies for carriers of haplogroup A (HD = 0.999) [9] in Chinese (HD = 0.952) [36], Indian (HD = 0.992) [37], Mongolian (HD = 0.997) [38], and Ethiopian local goats (HD = 0.950–1.000) [39]. The estimated average nucleotide diversity in our study was a bit lower in comparison with other studies, which were obtained in Chinese (π = 0.032) [40] and in Mongolian domestic goats (π = 0.0283) [38], and was close to the values estimated in Ethiopian local goats (π = 0.0093–0.0180) [39]. Russian goats had similar numbers of nucleotide differences compared to Ethiopian local goats [39]. Thus, the genetic and nucleotide diversity of Russian goats did not sway significantly from the values reported previously, which might indirectly indicate the utility of our calculation approach.

Analysis of the AMOVA results showed that genetic diversity mainly accounted for the within-breed differences, even within one haplogroup. Thus, inclusion in the analysis of haplotypes assigned to other haplogroups only led to a slight increase in within-breed diversity. Tarekegn et al. [39] reported that 59.11% of the genetic variation occurred within the two haplogroups in Ethiopian local breeds, while 40.89% was explained by genetic differences between haplogroups A and G. However, the frequency of the second haplogroup was higher than in our study.

Based on the nucleotide sequence of mtDNA, six highly divergent goat haplogroups were determined including A, B, C, D, F, and G [8,9,41,42], which correspond to the maternal and geographical origin. The predominant clustering of Russian goats with the carriers of haplogroup A, revealed in our study, was expected and corresponds well to the data previously obtained by Naderi et al. [9], Colli et al. [7], and Tarekegn et al. [39]. Thus, Naderi et al. [9] reported that 90% of the goats of the Old World, as well as all goats of the New World, had haplogroup A. Furthermore, Colli et al. [7] noted that haplogroup A is the most ancient maternal lineage, originating in Southeastern Anatolia (Turkey), near the goat domestication center. In addition, Doro et al. [14] supposed that the wide spread of the haplogroup A lineage might result from breeding success and separation of domesticated goats from wild ones.

We found that two goats from the Altai Mountain breed were clustered with samples carrying rare haplogroups G and D, as well as a Soviet Mohair individual, which joined the haplogroup C carrier. Goats representing haplogroup C are found both in Asia and Europe [9]. Because of their low frequency, maternal lineages G and D are less studied in comparison with A, B, and C.

The haplogroup G group was first identified by Naderi et al. [9] in goats of Middle Eastern and Northern African origin, while haplogroup D was found in the Asian and Northern European animals. According to Colli et al. [7], only the Iranian bezoars belonged to haplogroups D and G. In neighboring Kazakhstan, haplogroup A was the most frequent in local goat populations [17]. Haplogroups C and D were also found there, with frequencies similar to those calculated in our work (1.0% and 1.4%, respectively). In addition, low frequencies of haplogroups C and D were reported in Mongolian local goats [38]. Based on the data analysis and their own results, Tabata et al. [17] supposed that domestic goats spread into Central Asia through the Silk Road via the so-called Eurasian Steppe belt. The presence of haplogroup D (Hap-57) in the Altai Mountain breed, originated in the border area of Kazakhstan, and clustering with goats from countries, related to the Silk Road (Kyrgyzstan India), can be considered as supporting data for the Tabata et al. hypothesis [17].

Revealing a cluster of Altai Mountain individuals with goats carrying haplogroup G was more puzzling. Akis et al. [43] showed that haplogroup G was found in Angora and Anatolian Black breeds. Although the Angora breed was used for creation of the Altai Mountain breed [15,16], it is doubtful that Angora females were imported to the USSR. Thus, we may hypothesize that maternal lineage G was brought from the Middle East through another Silk Road route. This is further supported by our previous whole-genome study of local sheep that originated in Siberian and Altai regions, which revealed a shared genetic background between Russian local and Iranian sheep [44]. Nevertheless, considering the use of different marker types, such an assumption should be treated cautiously and needs to be more precisely investigated. Moreover, the phylogenetic studies based on maternal lineages will be continued on expanded sample sizes and with using the whole mitogenome sequencing data.

## 5. Conclusions

We provided the results of the first investigation of D-loop sequence variability in seven Russian goat populations. Along with expected predominant clustering of Russian goats with the carriers of haplogroup A, the closeness to three other haplogroups (C, D, G) might indicate different migration routes, which were used to bring the goats into the territory of modern Russia. In addition, based on our results, we may assume that haplotype identification might be considered as a useful tool for creating programs for the maintenance and preservation of local goats, which are reared in neighboring, partially overlapping habitats.

## Figures and Tables

**Figure 1 animals-10-01603-f001:**
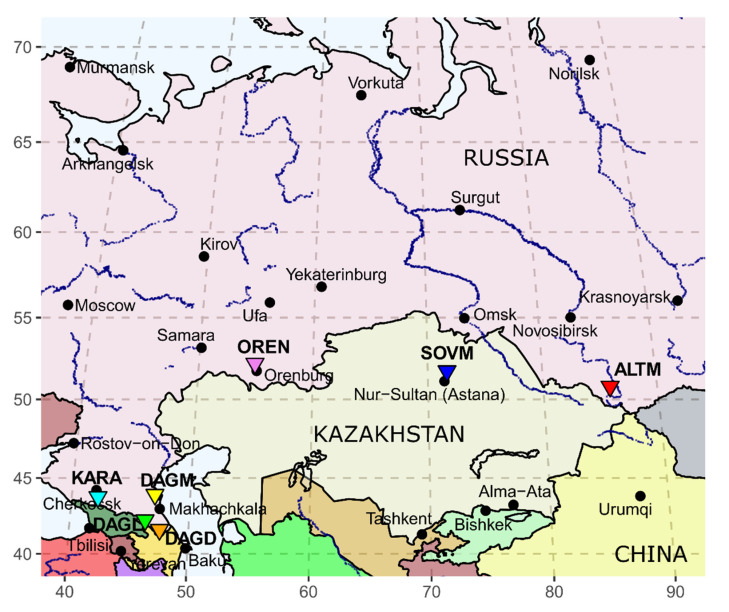
Geographical map showing the sites where seven studied goat populations originated. The goat groups are presented as inverted triangles colored in red for Altai Mountain (ALTM), in orange for Dagestan Downy (DAGD), in pink for Orenburg (OREN), in blue for Soviet Mohair (SOVM), in light green for Dagestan Local (DAGL), in yellow for Dagestan Milk (DAGM), and in turquoise for Karachaev (KARA) breeds.

**Figure 2 animals-10-01603-f002:**
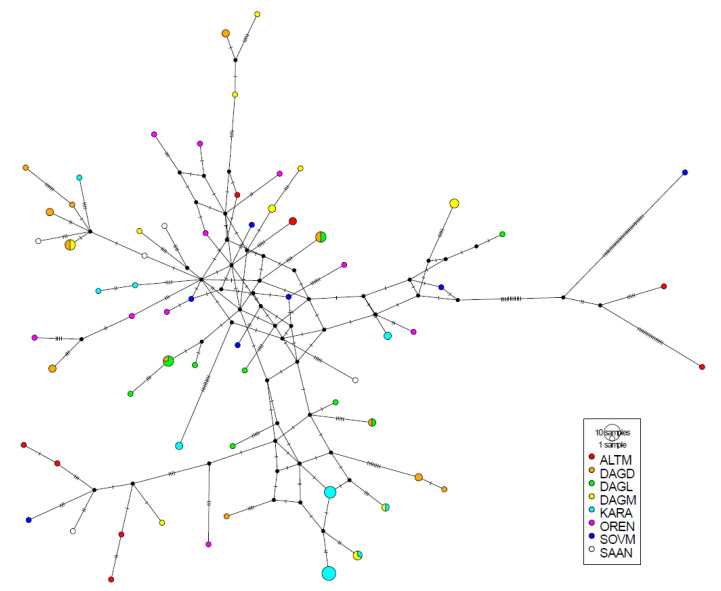
Median-joining networks for the 59 various haplotypes found in seven Russian goat populations and five haplotypes of the Saanen breed. Full names of breeds are shown in Table 1.

**Figure 3 animals-10-01603-f003:**
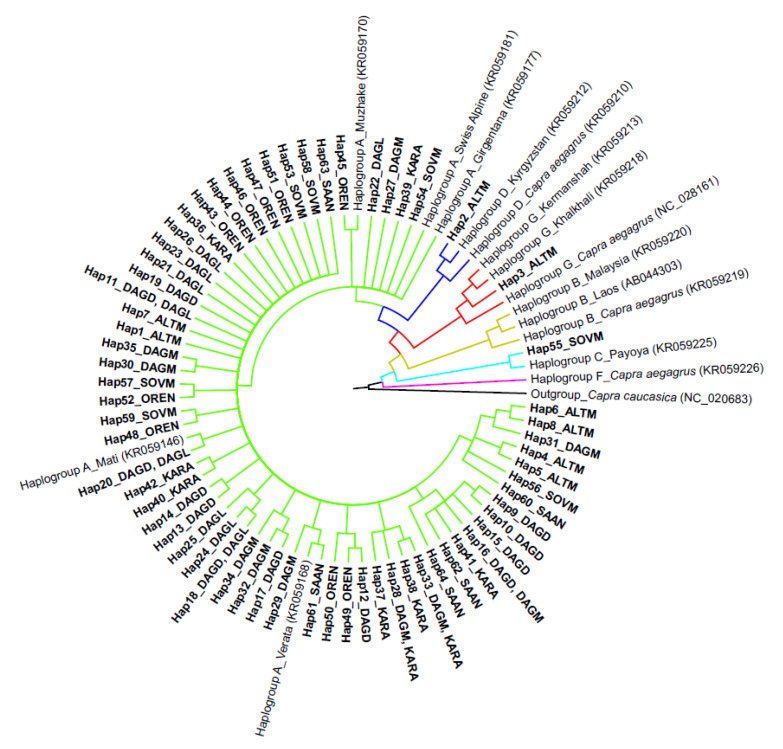
Bayesian phylogenetic tree based on 715 bp D-loop mtDNA sequences of seven Russian goat populations and reference haplotypes belonging to six goat haplogroups with Caucasian tur (Capra caucasica) as the outgroup. The following colors are used to highlight the haplogroups: green for haplogroup A, yellow-green for haplogroup B, turquoise for haplogroup C, blue for haplogroup D, pink for haplogroup F, and red for haplogroup G. Full names of breeds are shown in Table 1. Assignment of haplotypes to specific breeds is presented in Table 3.

**Table 1 animals-10-01603-t001:** Genetic diversity of mitochondrial DNA (mtDNA) in the goats reared in Russia.

Breed/Group	Code	N ^1^	S ^2^	H ^3^	HD ^4^	K ^5^	π ^6^
Altai Mountain	ALTM	9	45	8	0.972 ± 0.064	16.222	0.02266 ± 0.00466
Dagestan Downy	DAGD	18	45	12	0.961 ± 0.026	11.608	0.01619 ± 0.00137
Dagestan Local	DAGL	12	28	9	0.939 ± 0.058	7.364	0.01027 ± 0.00138
Dagestan Milk	DAGM	15	38	10	0.943 ± 0.040	10.781	0.01542 ± 0.00114
Karachaev	KARA	21	36	9	0.843 ± 0.057	7.848	0.01121 ± 0.00197
Orenburg	OREN	10	30	10	1.000 ± 0.045	8.844	0.01234 ± 0.00147
Soviet Mohair	SOVM	7	57	7	1.000 ± 0.076	18.667	0.02607 ± 0.01078
Saanen	SAAN	5	25	5	1.000 ± 0.126	10.600	0.01478 ± 0.00297

^1^ n—sample number; ^2^ S—number of variable sites; ^3^ H—number of haplotypes; ^4^ HD—haplotype diversity; ^5^ k—average number of nucleotide differences; ^6^ π—nucleotide diversity.

**Table 2 animals-10-01603-t002:** Results of analysis of molecular variance (AMOVA) based on the analysis of the D-loop in seven Russian goat populations.

Source of Variation	Russian Breeds				Russian Breeds Clustered with Samples Carrying Haplogroup A			
	d.f.	SS	VC	V%	d.f.	SS	VC	V%
Among breeds	6	194.954	1.99876	22.76	6	187.719	2.03601	24.98
Within breeds	85	576.611	6.78366	77.24	82	501.405	6.11469	75.02
Total	91	771.565	8.78242		88	689.124	8.15070	

Notes: d.f—degrees of freedom; SS—sum of squares; VC—variance components; V%—percent of variation.

**Table 3 animals-10-01603-t003:** Distribution of haplotypes among studied breeds.

Haplotype	Number of Variable Sites ^1^	Distribution of Haplotypes ^2^ among Studied Breeds								Haplotype	Number of Variable Sites ^1^	Distribution of Haplotypes ^3^ among Studied Breeds							
		Altai Mountain	Dagestan Downy	Dagestan Local	Dagestan Milk	Karachaev	Orenburg	Soviet Mohair	Saanen			Altai Mountain	Dagestan Downy	Dagestan Local	Dagestan Milk	Karachaev	Orenburg	Soviet Mohair	Saanen
H-1	5	2								H-33	8 ^4^				2	1			
H-2	22	1								H-34	8				1				
H-3	24 ^3^	1								H-35	6				1				
H-4	8	1								H-36	12					2			
H-5	9	1								H-37	5 ^4^					5			
H-6	9	1								H-38	8 ^4^					7			
H-7	3	1								H-39	7					2			
H-8	7	1								H-40	5					1			
H-9	10		1							H-41	8					1			
H-10	4		1							H-42	7					1			
H-11	6		2	2						H-43	7						1		
H-12	7		2							H-44	3						1		
H-13	14		1							H-45	8						1		
H-14	13		2							H-46	1						1		
H-15	5		2							H-47	6						1		
H-16	4		2		2					H-48	3						1		
H-17	8		2							H-49	8						1		
H-18	3		1	3						H-50	4						1		
H-19	7		1							H-51	3						1		
H-20	9		1	1						H-52	9						1		
H-21	5			1						H-53	3							1	
H-22	6			1						H-54	6							1	
H-23	6			1						H-55	46							1	
H-24	6			1						H-56	8 ^5^							1	
H-25	2			1						H-57	3							1	
H-26	8			1						H-58	3							1	
H-27	12				3					H-59	2							1	
H-28	9 ^4^				1	1				H-60	7								1
H-29	7				1					H-61	7								1
H-30	3				2					H-62	6								1
H-31	7 ^5^				1					H-63	8								1
H-32	6				1					H-64	2								1

^1^ The number of variable sites (nucleotide substitutions) are shown in comparison to the reference sequence (Gene Bank Accession Number NC_005044.2); ^2^ Number of animals that revealed the haplotype; ^3^ Additional deletion of one nucleotide in position 15,947 of NC_005044.2; ^4^ Additional deletion of 17 nucleotides (position from 15,613 to 15,629 of NC_005044.2); ^5^ Additional deletion of one nucleotide in position 15,707 of NC_005044.2.

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
