# Peer review of "Maternal Origins and Haplotype Diversity of Seven Russian Goat Populations Based on the D-loop Sequence Variability"

_animals, 2020, doi:10.3390/ani10091603_

Round 1
Reviewer 1 Report
The manuscript present very limited results with even less discussion. The results conform to what would be expected (in terms of Haplotypes) and does not explain any significant impact that this will have on the industry.
The discussion is limited to the haplotype results, and does not touch on any of the other reported results.
More comments on the attached paper.
Extensive English editing is necessary.

Author Response
Dear Reviewer,
thank you very much for your comments!
Please find our responses below.
Point 1: The manuscript present very limited results with even less discussion. The results conform to what would be expected (in terms of Haplotypes) and does not explain any significant impact that this will have on the industry. The discussion is limited to the haplotype results and does not touch on any of the other reported results.

Response 1: Thank you very much for your comments!
Our paper is planned as reporting of the first results on study of the D-loop mtDNA variability in Russian goat breeds. We agree that original manuscript included limited results and discussion. We expanded results section by adding more detailed description of estimated genetic diversity indicators (P4L168-174), the better representation of haplotype distribution (Table 3) and AMOVA results (Table 2, P5L184-188). In addition, we included values of standard error for parameters of genetic and nucleotide diversity in Table 1.
We improved discussion by adding discussion on the pattern of haplotype distribution in Russian goats (P8-9 L241-261), more detailed discussion estimated parameters of genetic and nucleotide diversity with the results reported earlier in other goat breeds (P9L267-2274) and a brief disputing on the AMOVA results (P10L275-280). We provided some ideas how identification of the haplotype in the local populations inhabiting the overlapping zones might be useful for their conservation and official recognition.
Point 2: More comments on the attached paper. Please explain the importance of the breeds that you investigated in the Intro.
Response 2: We improved the Introduction by focusing on the current state of goat resources in Russia, their brief history, and their importance (P1-3 L40-74, P3L81-86.).
Point 3: Give more detail. Which local breeds were crossed with which exotic breeds?
Response 2: We added the relevant information in P3L81-86 as following: « Russian goat breeds mainly originated from native nondescript goats crossed with males from high productive breeds [15, 16], which were Angora for Soviet Mohair, local Don for Altai Mountain, and Soviet Mohair for Dagestan Downy breed»
Point 4: Please include approval number for this study
Response 4: We added approval number as following Protocol 3/1 from 4th December 2019.
Point 5: What kind of tissue samples?
Response 5: There were ear pinches.
Point 6: I fail to see the relevance of this (regarding Table 1 in original submission).
Response 6: We removed Table 1 from the main text to Supplementary Materials (Table S1). We excluded several non-relevant sequences from the comparison studies and leaved only the relevant samples. We added the references to the sequences used for comparative studies.
Point 7: Please explain in more detail
Response 7: We explained more precisely estimated parameters of genetic and nucleotide diversity (P4-5 L162-173).
Point 8: This isn't an average, but a total. I have no idea where this value come from. Nor an average or a total? Not average or total??? Try to make the 2nd-4th column narrower, so that the breed names fit on one line in the first column.
Response 8: We corrected the table to put the breed names in one line. We improved Table 1 (in revised paper) and removed ambiguous row.
Point 9: Can you indicate which populations are shown here - which lines?
Response 9: We removed Figure 1 to Supplementary Materials. To make the results, presented on the Figure 1 clearer, we replaced the positions of the variable sites, which were defined according to the nucleotide position in the sequenced fragment to the positions in the reference sequence. Based on the results, presented in Fig. 1, to arrange the identified haplotypes to the specific breeds we included the additional table in the main text (Table 3), where we presented the distribution of the identified haplotypes between studied breeds.
Point 10: Will it be possible to group the haplotypes per breed? And indicate in different colours.
Response 10: We added Table 3 in which we grouped the haplotypes per breed.
Point 11: I suggest to include a Tree done by population (not individuals) for a clear picture re phylogenetics. The Figure is difficult to understand as there is no clear differentiation to indicate the various animals. The colours are not specific enough. Again, rather do this by population than by individual... Especially for HA
Response 11: Thank you for the suggestion. We suppose that the population tree would be not quite representative. According to the individual tree, presented on Fig. 3, we did not observe the clear clustering of the haplotypes related to the specific breeds. The haplotypes, which were identified in one breed, were clustered together with haplotypes, which were identified in other breeds. Because of the small sample size, the population tree will not reflect the relationship between studied breed. We reconstructed the individual tree, labeling the haplotypes related to different haplogroups by different colors. We improved Figure 2 and used more specific colors. We changed both MN and phylogenetic tree.
Point 12: Very limited Discussion, only on the Haplotypes. This need to be expanded and should show some deeper interpretation of the results.
Response 12: We improved discussion by adding discussion on the pattern of haplotype distribution in Russian goats (P8-9 L241-261), more detailed discussion estimated parameters of genetic and nucleotide diversity with the results reported earlier in other goat breeds (P9L267-2274) and a brief disputing on the AMOVA results (P10L275-280).
Point 13: Did you check that these weren't genotyping errors?.
Response 13: We checked thoroughly. The sequencing was performed in both directions in duplicates.
Point 14: Extensive English editing is necessary.
Response 14: We submitted the manuscript to English Editing Services of MDPI Author Services
Best regards,
Tatiana Deniskova

Reviewer 2 Report
In my view, this paper this paper presents important information about the evolution of the Russian goat local population. I think this paper can be accepted after a minor revision, specially centred in the discussion. This is completely descriptive, very few new ideas and explanations are supplied by the authors. It must be seriously improved to increase the interest of the document. The poor discussion also affect the conclusions, they are mostly repetitions of results reported in the corresponding section, authors must remark the importance of their findings to the science and to the productive sectors, for instance, the possible repercussion of these finding in the official recognitions of the breeds must be mentioned.
Also, the figures must be improved.
Some other questions must be considered.
Introduce “seven” goat populations…… in the title
In the abstract this is not a conclusion “In conclusions, our data provided the first insight into the phylogenetic pattern of Russian local goats as well as better understanding of spreading of domestic goats into Eurasia.” So, please remark here the main conclusion of the paper.
Why there are big differences in the sample sizes? Is it related to the population census? Samples are randomly chosen, or they have been collected in concrete farms?
Author Response
Dear Reviewer,
thank you very much for your comments!
Please find below our responses to the raised comments.
Point 1: In my view, this paper this paper presents important information about the evolution of the Russian goat local population. I think this paper can be accepted after a minor revision, specially centred in the discussion. This is completely descriptive, very few new ideas and explanations are supplied by the authors. It must be seriously improved to increase the interest of the document. The poor discussion also affect the conclusions, they are mostly repetitions of results reported in the corresponding section, authors must remark the importance of their findings to the science and to the productive sectors, for instance, the possible repercussion of these finding in the official recognitions of the breeds must be mentioned.

Response 1: Thank you very much for your comments!
We improved discussion by adding discussion on the pattern of haplotype distribution in Russian goats (P8-9 L241-261), more detailed discussion estimated parameters of genetic and nucleotide diversity with the results reported earlier in other goat breeds (P9L267-2274) and a brief disputing on the AMOVA results (P10L275-280). We provide some ideas how identification of the haplotype in the local populations inhabiting the overlapping zones might be useful for their conservation and official recognition.
Point 2: Also, the figures must be improved.
Response 2: We improved the figures and added the geographic map showing the sites where seven studied goat populations originated for better representation of the geo-evolutive relationships of the studied breeds for the readers. (Figure 1 in revised manuscript).
Point 3: Introduce “seven” goat populations…… in the title
Response 3: We corrected the title as it was suggested.
Point 4: In the abstract this is not a conclusion “In conclusions, our data provided the first insight into the phylogenetic pattern of Russian local goats as well as better understanding of spreading of domestic goats into Eurasia.” So, please remark here the main conclusion of the paper.
Response 4: We re-wrote the abstract and changed conclusions both in abstract and in the paper as following: « We provided the results of the first investigation of D-loop sequence variability in seven Russian goat populations. Along with expected predominant clustering of Russian goats with the carriers of haplogroup A, the closeness to three other haplogroups (C, D, G) might indicate different migration routes, which were used to bring the goats into the territory of modern Russia. In addition, based on our results, we may assume that haplotype identification might be considered as a useful tool for creating programs for the maintenance and preservation of local goats, which are reared in neighboring, partially overlapping habitats».
Point 5: Why there are big differences in the sample sizes? Is it related to the population census? Samples are randomly chosen, or they have been collected in concrete farms?
Response 5: For this study, we collected larger sample numbers from each population (from 20 to 25 samples) than the numbers, which were submitted to further sequencing of D-loop. This was due to there were a lot of highly-related animals in our initial collecting that would create possible biases in estimation of genetic diversity indicators. Thus, one animal from a pair of full sibs was left in our analysis. We stated this in P9L261 -266.
Sample sizes were not reflected the population census of the breeds. We added information on dynamics and current population census of the breeds in the Introduction to provide better understanding of the modern state of goat breeding in Russia (P1-2 L40-74).
Specimens of Orenburg, Altai Mountain, and Soviet Mohair breeds were collected in relevant officially registered breeding farms. Samples of Dagestan Downy, Karachaev, Dagestan Local, and Dagestan Milk groups were obtained from smallholders. Saanen samples were taken from breeding farm, which initially imported sires from France.
Best wishes,
Tatiana Deniskova

Reviewer 3 Report
This is an interesting paper about an unexplored matter, it is very well designed and redacted. Even the results are limited, and they have few international relevance. I think it has some new finding which could justify the publication in Animals.
In general, the paper needs some improvements in its projection. Authors must brave and open their mains to some speculations which could increase the output of the paper, presently in mostly descriptive. Also, the following remarks must be observed
It is not recommended to repeat the key words in the title
In the introduction must be remarked that the characterization is the first step in the conservation programs.
In material and methods, must be justified why the samples are unbalanced
Figures 1 and 3 must be improved, in their present form are not readable.
Discussion is very well redacted, but I think it must be enriched with more evolutive and geo-evolutive information, where are these population from?, which are their evolutive relationship?, among other questions must be answered more deeply.
References from other contexts are missed, authors must review the occidental influences on the Russian goat populations.
Author Response
Dear Reviewer,
thank you very much for your comments!
Please find our responses below.
Point 1: This is an interesting paper about an unexplored matter, it is very well designed and redacted. Even the results are limited, and they have few international relevance. I think it has some new finding which could justify the publication in Animals. In general, the paper needs some improvements in its projection. Authors must brave and open their mains to some speculations which could increase the output of the paper, presently in mostly descriptive.
Response 1: Thank you very much for your comments!
We improved discussion by adding discussion on the pattern of haplotype distribution in Russian goats (P8-9 L241-261), more detailed discussion estimated parameters of genetic and nucleotide diversity with the results reported earlier in other goat breeds (P9L267-2274) and a brief disputing on the AMOVA results (P10L275-280). We provide some ideas how identification of the haplotype in the local populations inhabiting the overlapping zones might be useful for their conservation and official recognition.
Point 2: It is not recommended to repeat the key words in the title
Response 2: We changed keywords to avoid repeating.
Point 3: In the introduction must be remarked that the characterization is the first step in the conservation programs.
Response 3: We added relevant remark into introduction (P3 L91-94)
Point 4: In material and methods, must be justified why the samples are unbalanced
Response 4: We added justification why the samples are unbalanced. For this study, we collected larger sample numbers from each population (from 20 to 25 samples) than the numbers, which were submitted to further sequencing of D-loop. This was due to there were a lot of highly-related animals in our initial collecting that would create possible biases in estimation of genetic diversity indicators. Thus, one animal from a pair of full sibs was left in our analysis. We stated this in P9L261 -266.
Point 5: Figures 1 and 3 must be improved, in their present form are not readable.
Response 5: We removed Figure 1 from main text into Supplementary Materials. To make the results, presented on the Figure 1 clearer, we replaced the positions of the variable sites, which were defined according to the nucleotide position in the sequenced fragment to the positions in the reference sequence. We remade the individual tree (Fig. 3), labelling the haplotypes related to different haplogroups by different colors.
Point 6: Discussion is very well redacted, but I think it must be enriched with more evolutive and geo-evolutive information, where are these population from?, which are their evolutive relationship?, among other questions must be answered more deeply.
Response 6: We addressed the geo-evolutive relationship of the studied goat breeds (P8-9 L24-258). We added the geographic map showing the sites where seven studied goat populations originated for better representation of the geo-evolutive relationships of the studied breeds for the readers. (Figure 1 in revised manuscript).
Point 7: References from other contexts are missed, authors must review the occidental influences on the Russian goat populations.
Response 7: We analyzed additional references. The influence of the Angora breed was addressed in P10 L305-308.
In another paper, which are currently under the review in other journal and based on SNP-data of Russian goats, we found the evidence of the occidental influences on the Russian goat populations, especially strong Saanen influence on Dagestan Milk population and traceable influence of Angora breed on Dagestan Downy and Soviet Mohair breeds. In addition, official recordings support these findings. Nevertheless, the relevant occidental influences were found based on using nuclear DNA markers while our data presented in this study do not support or exclude these influences.
We greatly appreciate your comment. However, the present data based on the D-loop sequences do not allow making such a strong hypothesis. As we are going to collect more samples and to perform whole mitogenome sequencing, we might find some evidence of the occidental influences based on maternal lineages.
Best regards,
Tatiana Deniskova

Round 2
Reviewer 1 Report
Dear authors
Thank you for the improvements and for addressing all comments.